# Heating Pre-Treatment of Copper Ores and Its Effects on the Bond Work Index

Nataly Cisternas [1] , Pablo Tobosque [2] , Daniel Sbarbaro [1,3,*] , Carlos Munnier [4] , Willy Kracht [4] and Claudia Carrasco [1,2]

1   Solar Energy Research Center, SERC, Av. Tupper 2007, Santiago 8370451, Chile; natalyban@gmail.com (N.C.); ccarrascoc@udec.cl (C.C.)
2   Department of Materials Engineering, Universidad de Concepción, Edmundo Larenas 270, Concepción 4070409, Chile; ptobosquepereira@gmail.com
3   Department of Electrical Engineering, Universidad de Concepción, Edmundo Larenas 219, Concepción 4070409, Chile
4   Department of Mining Engineering, Universidad de Chile, Tupper 2069, Santiago 8370451, Chile; cmunnier@udec.cl (C.M.); wkracht@uchile.cl (W.K.)
*   Correspondence: dsbarbar@udec.cl

**Abstract:** Comminution is the stage with the largest energy consumption in the mining process. Therefore, several pre-treatments have been proposed to reduce the energy requirements of this stage. This work analyzed the effect of a heating pre-treatment on the Bond index. A conventional heating pre-treatment was applied to a Chilean copper ore. The ore was heated to temperatures from 300 to 600 °C using a conventional furnace, resulting in a reduction of 19% in the Bond work index. Due to the pre-treatment, the mineral cracked in several areas. Microfracture and composition analyses of these areas confirmed that crack generation in the ore is due to the thermal stress produced by the pre-treatment. The fracture analysis explains the reduction in the Bond work index, since crack generation started at similar temperatures to those at which the reduction in the Bond work index was observed. In addition, the analysis also shows that micro-cracks occur between and through different phases, which may have an impact on mineral liberation. These results also show that, under a moderate high temperature, an important reduction in energy consumption can be obtained.

**Keywords:** bond work index; comminution; heating pre-treatment

## 1. Introduction

The mining industry is one of the cornerstones of Chilean economy. Nevertheless, the energy consumption related to this industry represents approximately 14% of the total Chilean energy consumption [1]. Therefore, the mining industry is one of the main sectors responsible for $CO_2$ emissions in the country. Comminution represents up to 50% of the total energy consumption in the mineral production process [2]. Moreover, comminution using ball milling is highly inefficient due to heat and mechanical losses, reaching efficiencies of only 2% [3]. The efficiency of comminution depends strongly on the hardness of the rock, which is constantly increasing due to the aging of mines and the greater depth of excavation [4].

Due to the enormous energy requirements of comminution, it is imperative to improve this process by implementing technologies that help to reduce its energy consumption. It has been previously shown that pre-treatments for the mineral can be implemented for this purpose [5,6], including microwave [7,8], ultrasonic [9], chemical [10], biological [11], and conventional heating pre-treatments [12]. The latter is based on thermal stress generated in the rock due to a change in temperature, which produces thermal expansion of the different compounds of the mineral determined by different expansion coefficients. Hence, it is well known that, by producing thermally induced stress, cracks can be induced in the rock [13].

The generation of cracks softens the rock, thus facilitating rock-size reduction, which in turns reduces the energy that is needed to process it. Additionally, it has been shown that heating pre-treatments improve mineral liberation in the last stages of production due to fracture generation [14,15].

Water cooling can provide further improvements in terms in energy consumption and liberation [16]. However, in this work, we were interested to demonstrate that, even without water cooling, it is possible to obtain an important reduction in the work index, since most of the main mining operations in Chile are in the desert, where water is a scarce and costly resource.

Previous studies showed that a thermal treatment that reaches temperatures above 400 °C can reduce the grindability of iron ore after 1 h of treatment [17] and induce microcracks in Laurentian granite [18]. It is important to note that fracture generation depends on the temperature and on its physicochemical characteristics [17]. Thus, it is very important to explore the effect of heating pre-treatment on copper ores found in specific regions to study the technical feasibility and find the optimal operational parameters. This is key information to enable the subsequent design and techno-economic analysis of possible pre-treatment strategies based, for example, on solar energy. In particular, it is important to mention that most mining industries are located in places where high levels of solar radiation are present [19]. Hence, it is possible to use solar energy instead of conventional energy sources during the pre-treatment, and thus reduce $CO_2$ emissions in a solar-based manner. This will not only have an economical benefit, but also an environmental one. In this context, this work represents an initial study necessary to fulfill a long-term goal of using direct solar energy for improving the comminution processes.

Most of the works reported in the literature provide information about the effect of pre-heating on structural changes at microscopic scales. However, to the best of our knowledge, there are no studies on grindability indices such as the Bond index. Thus, this work addresses the problem of quantifying the effect of heat pre-treatment on the Bond index, specifically for a Chilean copper ore. However, the methodology used in this work can also be applied to the research of any ore around the world.

This paper is organized as follows: Section 2 describes the materials and methods. Section 3 summarize the experimental results and their analysis. Finally, in Section 4 some conclusions and further work are outlined.

## 2. Materials and Methods

More than 55 samples from a mafic complex [20] were collected from a set of representative drill cores. Several sections having a 5 cm diameter are shown in Figure 1. They have predominant porphyry and interstitial textures, with contents of phenocrystals between 5% and 30%, and silica between 47.7% and 54.7%. The X-ray diffractogram, in Figure 2, shows the presence of quartz, biotite, tourmaline, chalcopyrite, and anhydrite.

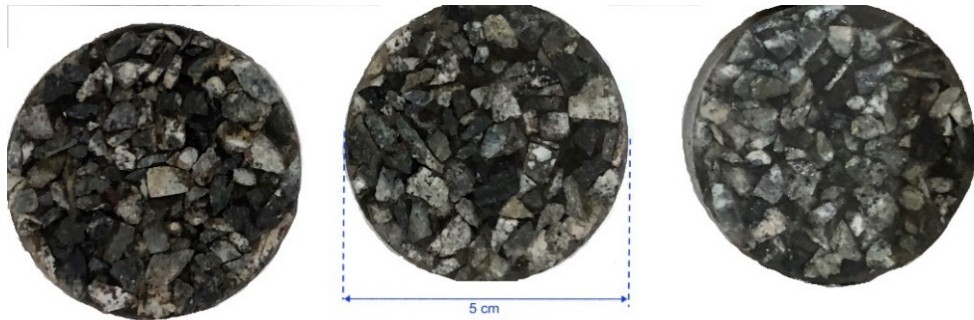

**Figure 1.** Images of some copper ore samples used in the study.

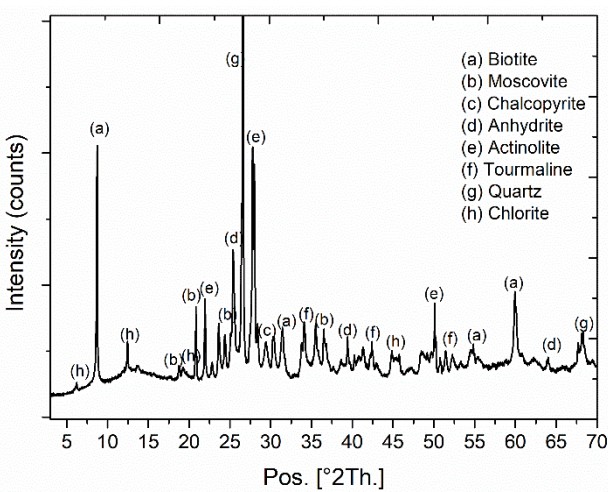

**Figure 2.** X-ray diffractograms and the main mineral component of ore samples.

Samples of 7 kg were used, having a size-# 6 (3.360 μm). Samples were heated in a custom-made electric furnace with a volume of 9.4 L and a rated power of 4.4 kW at temperatures of 300, 400, 500, and 600 °C. The furnace reached the desired temperature in about 1 h and the mineral was left inside for 1.5 h. The heating was carried out in an oxidizing atmosphere (air) and the sample was placed in a stainless steel container. At 600 °C, some samples presented some oxidation. The mineral inside this container was stirred manually and constantly. Therefore, the temperature of the mineral particles that were in contact with the wall of the container was not significantly different from the temperature of the particles that were at the center of the batch. The temperature of the mineral was measured in situ by means of a K-type thermocouple. For each heating range, an error of ±10 °C was defined from the final temperature established. After the time elapsed, each 7 kg sample was allowed to cool in air at room temperature. These samples were separated into ten pieces of 700 g portions using a rotary divider to ensure homogeneity of each portion. This procedure was repeated twice for each temperature with different samples.

Grindability tests were carried out through the conventional Bond method [21] using a balls Bond mill EDEMET of size 305 × 305 mm with rounded edges. The mill had 20.125 kg of steel balls with the size distribution given in Table 1. Using this mill, it was possible to determine the Bond work index and energy required for the size-reduction process. The Bond work index and energy are given by the following formulas, respectively:

$$W_i = \frac{44.5}{P_1{}^{0.23} GPR^{0.82} \left( \frac{10}{\sqrt{P}} - \frac{10}{\sqrt{F}} \right)}, \tag{1}$$

$$E = W_i \left[ \frac{10}{\sqrt{P}} - \frac{10}{\sqrt{F}} \right], \tag{2}$$

where $P_1$ is the 100% passing size of product in microns (3.350 mm), $GPR$ is the grams per revolution, $P$ is the 80% passing size of the product in microns, and $F$ is the passing size of feed in microns (∼ 2000 μm). $W_i$ and $E$ are in units of kWh/ton. The values of $P$ and $F$ were calculated by a granulometric analysis using a RX29 Ro-Tap Sieve Shaker.

**Table 1.** Distribution of steel balls for the Bond test mill.

| Quantity of Balls | Size (mm) |
|---|---|
| 43 | 36.8 |
| 67 | 29.72 |
| 10 | 25.40 |
| 71 | 19.05 |
| 94 | 15.49 |

A conventional Bond test was performed each time for 700 g of mineral previously exposed to several temperatures from 300 to 600 °C, and for un-treated mineral.

In addition to the Bond test, the mineral was analyzed in terms of fracture generation due to the thermal stress produced by the heating pre-treatment. In this case, samples of approximately $3 \times 3 \times 1$ cm$^3$ were heated in an electric furnace. Two sets of mineral samples were heated to temperatures of 200, 300, 400, 500, 600, 700, 800, 900, and 1000 °C and maintained at such temperature for 30 min., so a homogenous temperature could be reached inside each sample. After this 30 min. period, samples were removed from the furnace and cooled in air until they reached room temperature.

The samples were analyzed using a Olympus SZ61 stereoscopic microscope (Olympus Corporation, Tokyo, Japan). Further analysis was carried out using a Scanning Electron Microscopy (SEM) model JEOL JSM-6380 instrument equipped with an incorporated Energy Dispersive Spectroscope (EDS) (Jeol Ltd., Tokyo, Japan). The mineralogy characterization was performed using X-ray diffraction in a Bruker AXS D500 diffractometer with Cu-K$\alpha$ radiation (Bruker, Berlin, Germany). Additionally, to determine possible phase transformations, Differential Scanning Calorimetry (DSC) was carried out using a Netzsch DSC 404F3 device (NETZSCH-Gerätebau GmbH, Selb, Germany). Samples were analyzed in duplicate in dynamic mode, with a heating rate of 10 K/min and air atmosphere. No baseline adjustment was made. This analysis was also used to estimate the specific heat capacity of the samples. Since these measurements can be affected by exo/endothermic reactions, the estimate is an apparent specific heat capacity.

## 3. Results

### 3.1. Bond Tests

The results of the work index calculated according to Equation (1) as a function of the pre-treatment temperature are shown in Figure 3. For comparison, the data point shown at 0 °C indicates the work index of the untreated mineral. From the figure, it is possible to see that the important decrease in the work index starts only when the mineral was previously heated to 500 °C. Before this temperature, small variations are also observed. From the plot, it is possible to see that the work index had a reduction of 19% when the mineral was previously heated to 600 °C. These results are consistent with those reported in the literature for other minerals, where the reduction in the Bond work index was reduced after an exposure to temperatures above 400 °C depending on the mineral composition [16–18]. As an example, in the work undertaken by Omram et al. [22], the grindability of iron ore increased from 46.6 to 50.8% after a thermal treatment for 1 h up to 600 °C. In the case presented here, the grindability started to increase even for a treatment that exposed the mineral up to 400 °C for 1 h.

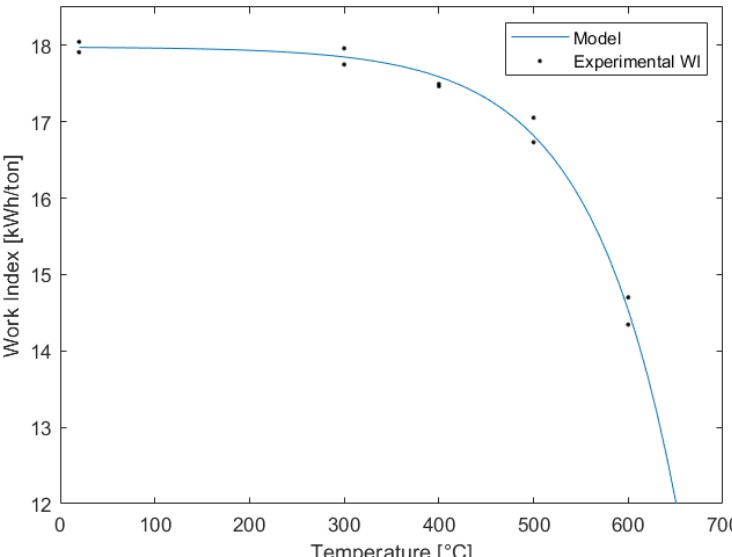

**Figure 3.** Work index as a function of the pre-treatment temperature.

Copper ore is composed mostly of quartz, which is consistent with the results shown in Figure 3. Since the temperature of the first allotropic transformation of quartz is 573 °C [23], at this temperature, a considerable increase in volume occurs, which is very likely to cause fracture generation in the mineral, explaining the decrease in the work index.

An empirical model with the following structure

$$W_i = c\left(1 - e^{b(T-a)}\right),\tag{3}$$

was fitted to the data. The values of the parameters $a$, $b$, and $c$ were obtained by minimizing the squared mean error. The values of the adjusted parameters were $a = 751.7$, $b = 0.01089$, and $c = 17.98$, with $R^2 = 0.9888$.

In Figure 4a, the grindability of the minerals is shown as a function of the pre-treatment temperature along with the fitted model. The model in this case has the same structure given by Equation (3), but with a positive exponential term. The parameters in this case are $a = 758.4$, $b = 0.008903$, and $c = 1.06$, with $R^2 = 0.98838$. From the plot, it is possible to see an increase in grindability; this means that, after the treatment, fewer revolutions per gram would be required to reduce its size. Therefore, less processing would be needed to obtain the same results, which has as an economic benefit as a consequence. In the case of the analyzed copper ore, the grams per revolution increased by a 23%. It is important to mention that, as the grindability of the minerals is improved, other benefits are also achieved. The mills that are used for the process will have less wear, as less working time will be needed to reduce the size of the mineral. As the residence time of the mineral in the milling circuits would be reduced, it can have a large economic benefit for the industry since the time of processing is decreased. Moreover, another advantage of reducing the processing time is that acoustic contamination per ton of treated ore will be also reduced.

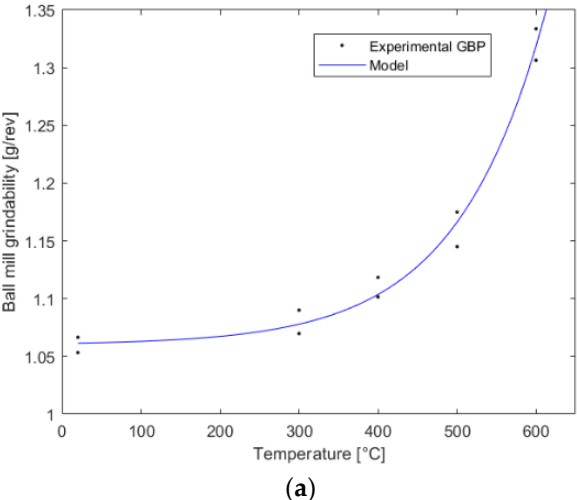
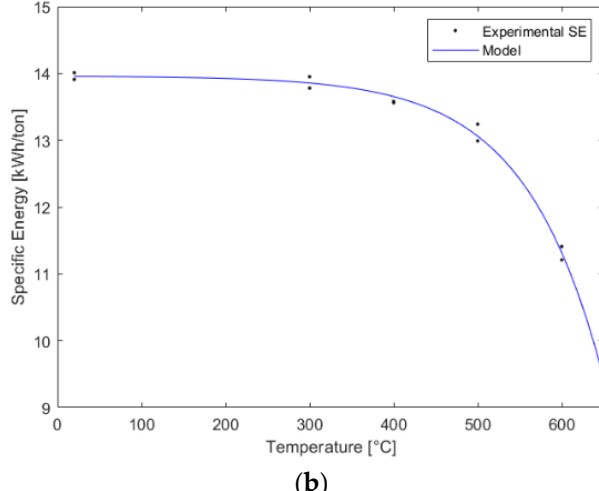

**Figure 4.** Grindability and specific energy as a function of the pre-treatment temperature: (**a**) ball mill grindability; (**b**) specific energy.

The energy needed to reduce the size of the mineral from an 80% passing size of *F* to an 80% passing size of *P* as a function of the pre-treatment temperature is shown in Figure 4b. In the same figure, a fitted model having the same structure given by Equation (3) is also displayed.

The value of the parameters obtained from the model were $a = 754.4$, $b = 0.01078$, and $c = 13.96$, with $R^2 = 0.9904$. From this figure, it is possible to see that the energy is reduced even for temperatures up to 400 °C. Actually, the energy is reduced by 8% when the ore is heated to 400 °C, which is a considerable amount of energy reduction considering the total amount of energy used in the mining industry. Moreover, as seen in the figure, the pre-treatment has significant implications in terms of energy, since the energy required to process the mineral is reduced by 16% when the pre-treatment reaches a temperature of 600 °C.

Given the positive effect of the pre-treatment on the mineral grindability, a more detailed microstructural analysis was carried out to investigate the effects of thermally stressing the ore.

*3.2. Microstructural Analysis*

As previously mentioned, ore samples were thermally treated at different temperatures to observe in detail the fracture generation produced by thermal stress in the mineral. Images of thermally treated ores subjected to different temperatures are shown in Figure 5. From the images, it is possible to see that fracture generation starts at a large scale for temperatures above 600 °C. In samples heated at higher temperatures, in Figure 5c,d, large cracks are observed of up to several millimeters, which can be indicative of a large thermal stress that can be attributed to a change in the volume of the components of the mineral. The change in volume can be attributed either to a phase transformation or to differences in the thermal expansion coefficients of the components of the mineral. It is important to mention that the fractures were encountered throughout the entire surface of the sample. Moreover, the crack density is seemingly larger for samples treated to temperatures above 600 °C.

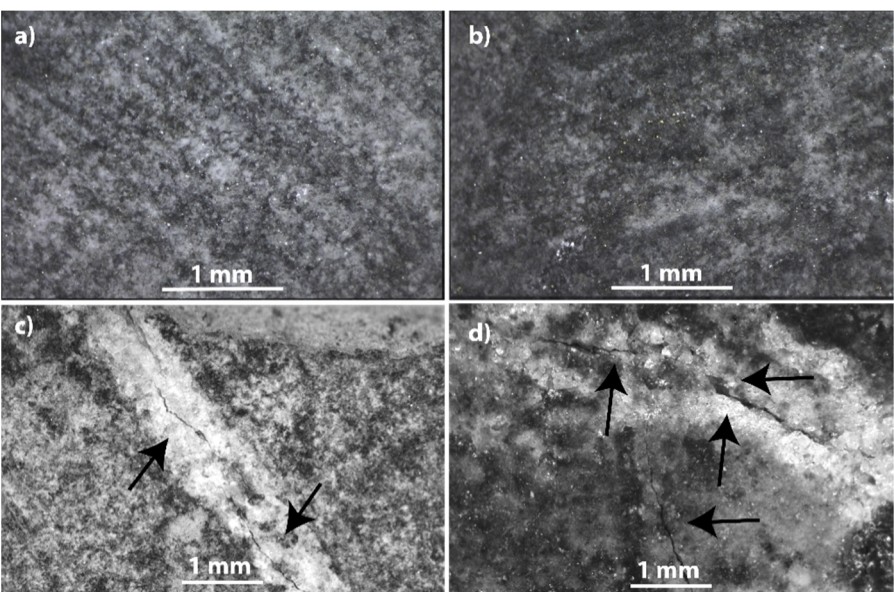

**Figure 5.** Images of the mineral thermally treated to temperatures of (**a**) 400 °C, (**b**) 600 °C, (**c**) 800 °C, and (**d**) 1000 °C.

As has been previously shown [24], crack generation seems to start at the edges of the samples and propagates, where weaker fracture branches are also observed. This behavior can be observed in Figure 6, where several images of the sample treated up to 1000 °C are shown. In these images, cracks of various sizes are shown.

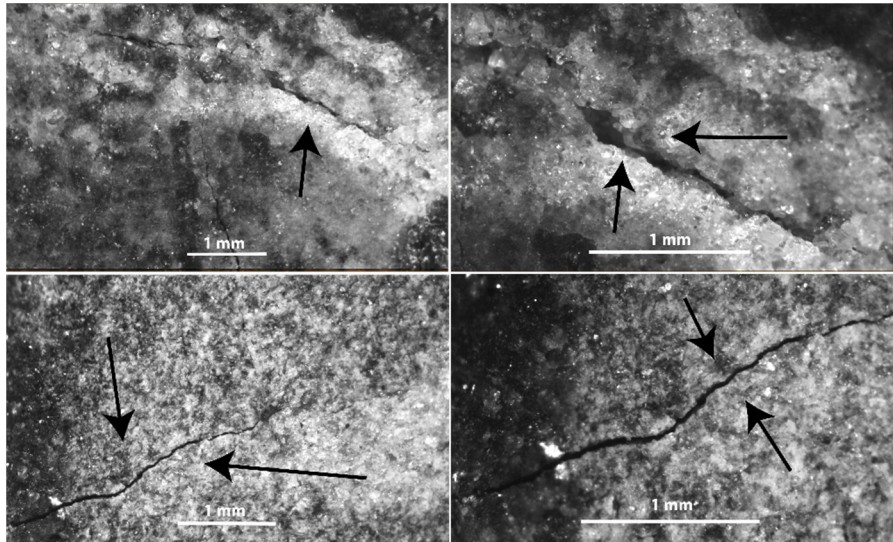

**Figure 6.** Fracture generation in samples thermally treated to 1000 °C.

According to the energy analysis, fracture generation should start at around 300 °C, where the Bond work index and the energy needed to reduce the size of the rock are reduced (see Figure 3). This is not observed in the pictures shown in Figures 5 and 6, likely because of the scale at which these images were taken. For a more in-depth observation of cracks, SEM images of the cracked samples previously treated at different temperatures are shown in Figure 7.

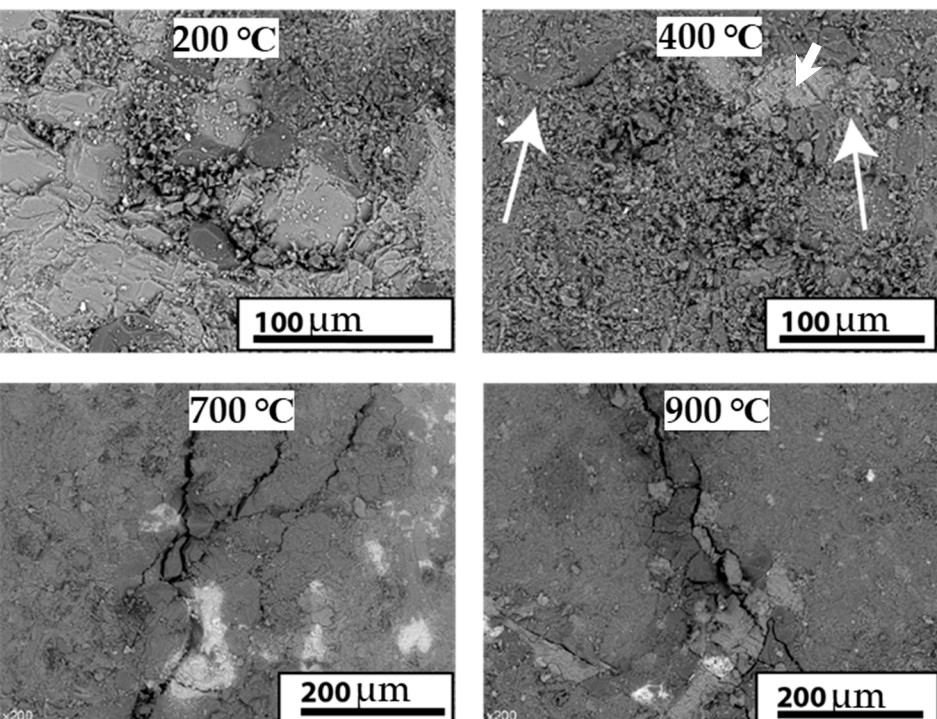

**Figure 7.** SEM images of mineral heated up to temperatures from 200 to 900 °C.

In Figure 7, SEM images are shown. In these images it is possible to observe that cracks appear clearly above 400 °C. Below this temperature, fractures can also be observed; however, the fracture is not as large as for the samples heated above 400 °C. From the images, it is also possible to see that crack propagation occurs between and through different phases. This is clearly observed in one of the images of the sample heated to 700 °C and the image showing the sample heated to 900 °C. Therefore, from the SEM images, it is possible to infer that crack formation occurs due to differences in the thermal expansion coefficients of the components of the mineral. It is important to note that, for lower temperatures (below 500 °C), intergranular fractures are observed, whereas for higher temperatures (above 500 °C), both intergranular and transgranular fractures are observed.

In order to determine the mechanism behind crack formation, XRD analyses were performed. XRD analyses also help to identify any phase transformations that may have happened during the heating of the samples.

Figure 8 shows the diffractograms of the ore samples at room temperature and heated to 400, 500, and 900 °C. For comparison, the diffractograms are shown with a vertical displacement. As can be seen in the figure, no significant differences in the diffractograms of the respective samples were observed. The phases identified correspond to biotite (JCPDS card no. 042-0603), quartz (JCPDS card no. 046-1045), chalcopyrite (JCPDS card no. 035-0752), muscovite (JCPDS card no. 01-086-1368), anhydrite (JCPDS card no. 37-1496), actinolite (JCPDS card no. 01-080-0521), tourmaline (JCPDS card no. 011-0592), and chlorite (JCPDS card no. 002-0012). These results show that no phase transformations occur during the heating to 900 °C. However, a deeper analysis made by DSC showed some interesting results. Figure 9 shows the DSC curve for the mineral under study in the range from 20 to 600 °C.

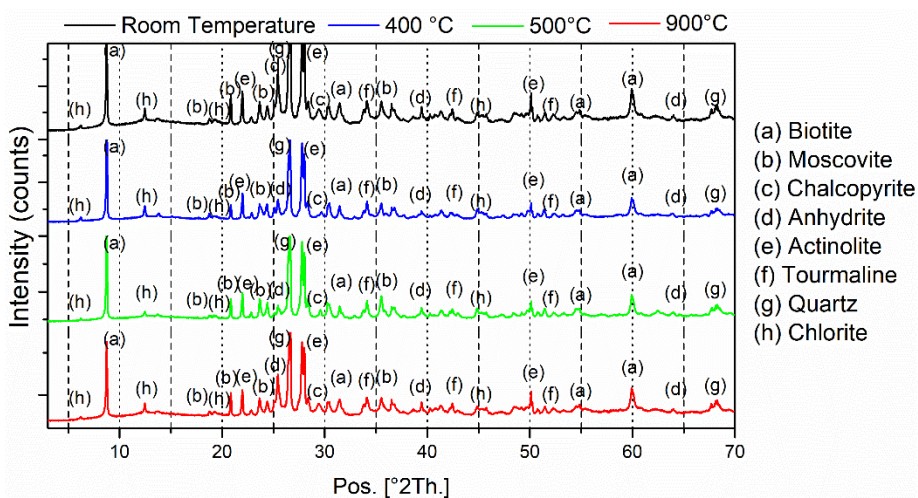

**Figure 8.** X-ray diffractograms of ore samples at room temperature and heated to 400, 500, and 900 °C.

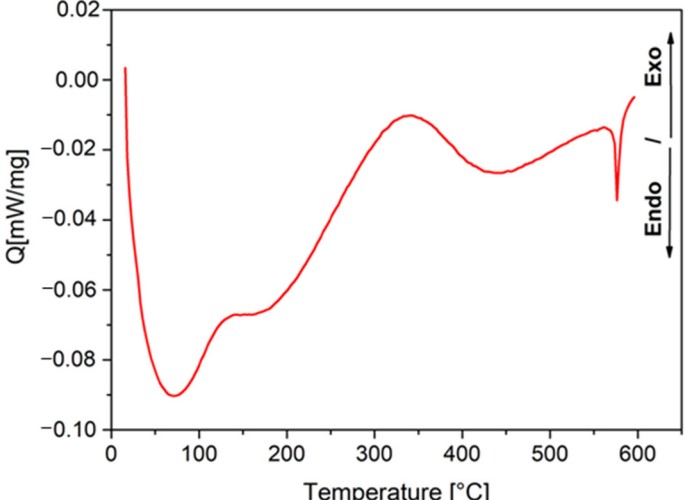

**Figure 9.** DSC curve for copper ore.

From Figure 9, it is possible to observe three main exothermic peaks after the initial endothermic hook. The first, at around 150 °C, is normally attributed to moisture loss. The second, located around 340 °C, has been explained as the decomposition of phases containing carbon and sulfur [25], in addition to hydroxides or nitrates [26]. In the studied mineral, and according to the literature, the exothermic peak between 300 and 400 °C can be attributed to changes in chalcopyrite, which, at around 360 °C, forms iron sulfate and results in the oxidation of sulfur [27]. In the same way, Carvalho et al. reported the crystalline modification of the anhydrite at around 380 °C [28], which cannot be excluded. Finally, the peak at around 570 °C can be associated with the silica leading to quartz inversion, which is normally reported at 573 °C [26].

To elucidate the mechanism behind the formation of cracks, it is necessary to identify the phases that were crossed by cracks. Mapping of the distribution of the elements was undertaken using EDS in a cracking zone of the sample heated to 900 °C. In Figure 10, it is possible to see the SEM image of the cracked zone, in addition to the distribution of S, Ca, Si, Ti, K, Al, Mg, Na, and Fe. In the analysis, copper, oxygen, and carbon were also detected; however, they are not shown here because the signal was within the detection limit of the device. In particular, copper was not easily detected because the ore grade was very low.

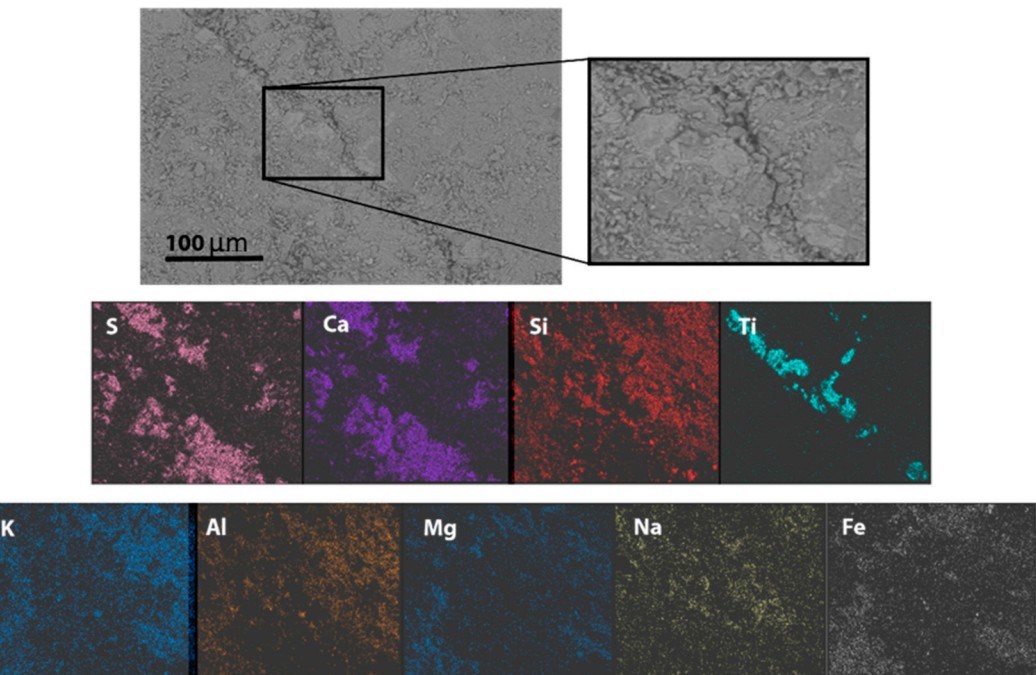

**Figure 10.** Mapping analysis of the sample heated to 900 °C.

From the analysis, it is possible to identify that cracks appeared mostly through the main crystalline phase in the sample corresponding to quartz (Si—red) and biotite (clearly Ti—calypso, although S—orange, Si—red, and Mg-K—blue can also be part of the mineral); whereas phases such as tourmaline (Fe—grey, Na—yellow, among others) and anhydrite (Ca—purple, S—pink, and others) did not show the presence of cracks. Thus, it is likely that the mechanism of crack formation is related to the phase transformations; at lower temperatures (between 300 and 400 °C), the decomposition of chalcopyrite can be responsible for the first cracks formed during the heating of the mineral. As the ore grade is very low, small quantities of cracks are expected, which is in agreement with the low diminution of the energy needed for milling when the ore was heated at temperatures lower than 400 °C. In the same way, the abrupt drop in the Bond index for mineral heating up to 600 °C can be directly related to the volumetric expansion of quartz. However, note that this does not mean that the other phases are not involved in crack formation. This is not inconsistent with what was found through the XRD analyses, since there were small phase transformations and, thus, they could not be detected by XRD.

### 3.3. Aparent Specific Heat Capacity

In this section, a preliminary analysis of the required energy required to heat the sample is carried out based on the heat flows obtained by DSC. The specific heat capacity at constant pressure can be calculated as follows:

$$c_p = \frac{1}{m}\frac{\Delta Q}{\Delta T} = \frac{1}{m}\frac{\Delta Q/\Delta t}{\Delta T/\Delta t} ,\tag{4}$$

where $\Delta Q/\Delta t$ is the heat flux, $\Delta T/\Delta t$ is the heating rate, and $m$ is the sample mass. The specific energy exchanged by the sample when the temperature change from $T_1$ to $T_2$ is then obtained as:

$$E = \int_{T_1}^{T_2} c_p(T)dT.\tag{5}$$

Using the DSC data depicted in Figure 9, the apparent specific heat capacity was calculated in terms of the temperature, as given in Figure 11. The estimation of the energy required to heat the sample was carried out by the numeral integration of $c_p(T)$ using a

trapezoidal method. The estimated value for the specific energy exchange by the sample is $E = 37.48$ kWh/Ton. This value is high compared to the work index of the mineral when no pre-treatment is applied (see Figure 3). Therefore, it is important to propose a system in which the investment in energy can be properly justified. This can be based on a solar pre-treatment since the mining industry in Chile is mostly located in areas where high rates of solar radiation are found.

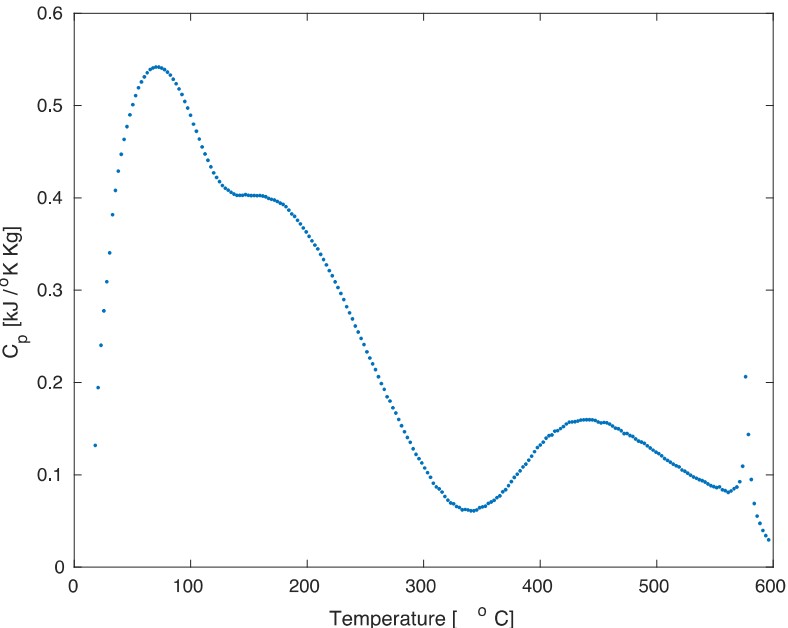

**Figure 11.** Apparent specific heat capacity.

A more in-depth techno-economic analysis of the implementation of a solar pre-treatment is outside the scope of this paper. This work focused on studying the effect of pre-heating copper ore to reduce its Bond index value. Future work will be focused on the balance of energy by also considering the use of solar energy as a source of heating.

### 3.4. Potential Economic and Operational Impacts

The reduction in the work index can impact the economic and operational characteristic of a grinding circuit. To study the different scenarios, Moly-Cop Tools 3.0 [29] was used to estimate the capacity and energy consumption of the grinding circuits.

To carry out the economic evaluation, the example data of the Sewell grinding plant were analyzed. The mill dimensions, number of mills, tonnage, and work index are available from [30]. This plant processes ores similar to those considered in this study. The first scenario analyzes the impact of a reduction in the work index on the processed tonnage.

Considering an F80 of 1250 microns and a total of 28 ball mills, the behavior of the total daily tonnage of the plant was calculated by varying the value of the work index associated with the different pre-treatment temperatures. The parameters of the simulations required by Moly-Cop Tools are summarized in Figure 12.

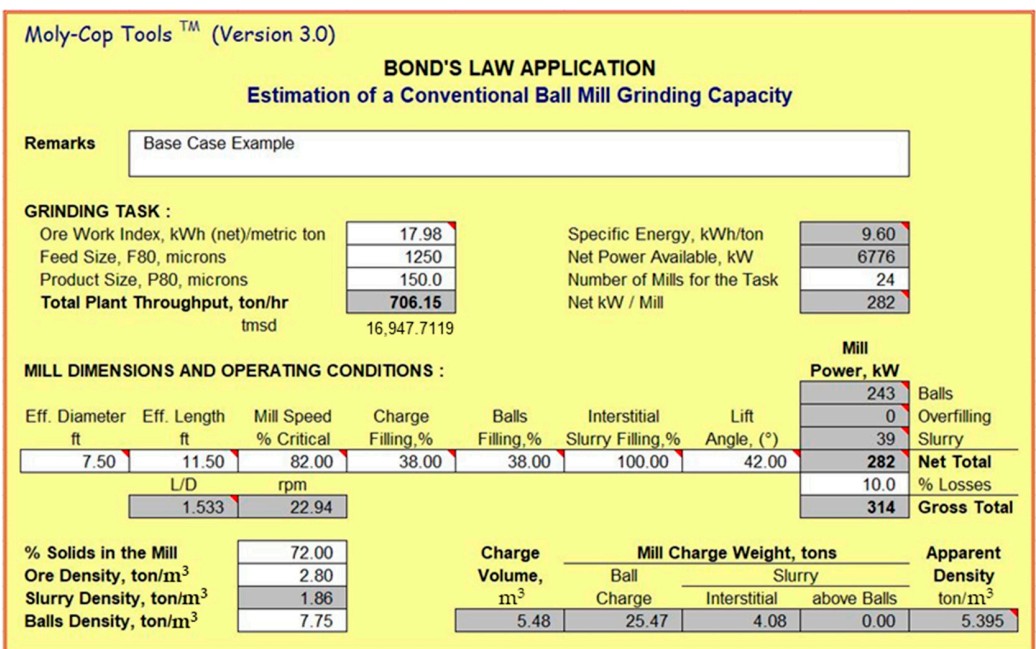

**Figure 12.** Simulation parameters: mill dimensions and operating conditions.

The incremental income is due to the increase in processed tonnage. The parameters used to calculate income are: recovery of 82% [30], ore grade of 0.685% [30], and copper price of 425.402 cUSD/lb [31]. Table 2 summarizes the results for the different work indices. Notice that the increment in tonnage is highly nonlinear.

**Table 2.** Incremental income due to a reduction in work index.

| Work Index (kWh/ton) | Tonnage (tmsd) | Difference Copper Ton/Day | Incremental Income (USD/Day) |
|---|---|---|---|
| 17.98 | 16,947.63 | - | - |
| 17.85 | 17,071.22 | 0.6943 | 6499 |
| 17.48 | 17,432.41 | 2.7231 | 25,530 |
| 16.89 | 18,041.52 | 6.1445 | 57,600 |
| 14.52 | 20,986.32 | 22.6855 | 212,684 |

The cost of heating 20,986 tons of CMET ore per day to 600 °C should be less than USD 212,554; for the pre-treatment to begin to be beneficial in economic terms, this is equivalent to heating 706.15 tons of ore per hour at a cost of USD 3605 per hour. Considering an energy cost of 110 USD/MWh [32], the net benefits will be 126,162 USD/day. It is important to remark that this increased tonnage cannot be processed in the actual grinding circuit since it would overrun its designed capacity.

By comparison, the second scenario considers the effect of the work index reduction on energy saving. The saved energy for grinding CMET ore with a 600 °C pre-treatment is of 2.12 kWh/ton, i.e., the difference between the work index at 25 and 600 °C. Per the DSC analysis, the energy value necessary to raise the temperature of the sample from 25 to 600 °C was 37.48 kWh/ton of ore. Thus, the use of electric energy for heating is not economically feasible.

In addition, it should be mentioned that, in this study, the effects of pre-treatment on ore liberation were not determined. Hence, there may also be improvements available in recovery and the final concentrate grade, which would lead to increments in income.

## 4. Conclusions

An important reduction of 19% in the Bond work index for a mafic complex copper ore can be obtained by performing heating pre-treatment. The microscopic analysis shows that this reduction is due to induced micro-cracks, which occur between and through different phases, and may have an impact on mineral liberation.

This study was the first of its kind performed with Chilean copper ores, and it is an important initial step in the development of a technology that fulfills the requirements of decreasing the energy consumption of the mining industry. Further work should consider analysis of different copper ores and carry out a techno-economic analysis of the feasibility of using solar energy as a heating source.

**Author Contributions:** Conceptualization, W.K., C.C. and D.S.; methodology, C.C.; software, P.T. and C.M.; validation, C.M., P.T. and N.C.; formal analysis, C.C.; investigation, C.M., N.C. and P.T.; resources, W.K. and C.C.; data curation, C.M.; writing—original draft preparation, N.C.; writing— review and editing, D.S., C.M., P.T. and C.C.; visualization, P.T. and C.M.; supervision, W.K., D.S. and C.C.; project administration, D.S.; funding acquisition, W.K. All authors have read and agreed to the published version of the manuscript.

**Funding:** This research was funded by the Solar Energy Research Center (SERC) FONDAP project grant number 15110019. P.T. was funded by ANID-PCHA/National Ph.D./2018-21181765.

**Conflicts of Interest:** The authors declare no conflict of interest.

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
