# Peer review of "Heating Pre-Treatment of Copper Ores and Its Effects on the Bond Work Index"

_minerals, doi:10.3390/min12050593_

Round 1

Reviewer 1 Report

It is very known that pre-treatment such as microwave, ultrasonic, chemical, biological or conventional heating of metallic ores before any comminution process creates cracks in the ores providing some benefits like decreasing the energy for crushing or grinding process. There are many studies in the literature. I did not see any innovation in this study except for repeating similar experimental works for another ore in this case is a Chilean copper ore. The current study is clearly in a dilemma about how to adapt this pre-treatment process in the copper industry since there was no solid conclusion whether these findings provide an economic benefit to the copper industry by means of electric consumption or not. The study needs a serious economic analysis base on technical feasibility given here as the authors mentioned in the discussion section saying that the works requires a further study. Therefore, I do recommend them to re-submit a new paper combining technical, i.e. experimental work given here, and economic study involving solar energy.  I also enclosed the revised paper.        

Reviewer 2 Report

This paper addresses the problem of quantifying the effect of heat-pretreatment on Bond index specifically for a Chilean copper ore. It shows good reduction in grinding index. It’s very important find way to improve the grind-ability of ore. However, for a good expiation of mechanism, the mineralogy of Chilean copper ore and coefficient of thermal expansion of individual minerals should be added. In addition, they should consider the following:

  1. line 68, 2. Materials and Methods

How about the content of quartz, biotite, tourmaline, chalcopyrite and anhydrite ?

What is the volume of the furnace?

What is the power of the furnace?

What about the particle size of sample?

how long did the furnace reach desired temperature? Did sample always in the furnace?

  1. How water cooling?

As I know, the below papers shows that water cooling have good effect on grinding.

The influence of mineralogy on microwave assisted grinding. Min. Eng. 2000, 13 (3): 313– 327.

Thermally Assisted Grinding of Cassiterite Associated with Pollimetallic Ore: A Comparison between Microwave and Conventional Furnaces. Minerals, 2021, 11(7):768.

Applications of microwave radiation to enhance performance of mineral separation processes. Richard Mozley Symposium, Falmouth, England. 1997.

Effecst of microwave radiation upon the mineralogy and magnetic processing of massive norweglan ilmenite ore. Magnetic and Electrical Separation, 1998, 9:131-148.

  1. For Bond test, the time was 1.5h, but for thermal stress analyzing, the time was 30 min. Does it have any effect?
  2. line 80,81, The heating was carried out in an oxidizing atmosphere (air). Is the sample oxidized badly?
  3. line 302, Discussion. It should conclusion, not discussion. In the paper, it did not used solar energy, why mention it in this part. To reduce CO2, but release SO2 due to oxidation.
  4. A total consumption energy should be discussion.

Round 2

Reviewer 1 Report

It seems ok, now. 

Reviewer 2 Report

 I think the manuscript has been sufficiently improved to warrant publication in Minerals.